# First Report of Two Cases of Löfgren’s Syndrome after SARS-CoV-2 Vaccination-Coincidence or Causality?

**DOI:** 10.3390/vaccines9111313

**Published:** 2021-11-11

**Authors:** Jan-Gerd Rademacher, Björn Tampe, Peter Korsten

**Affiliations:** Department of Nephrology and Rheumatology, University Medical Center Göttingen, 37075 Göttingen, Germany; jan-gerd.rademacher@med.uni-goettingen.de (J.-G.R.); bjoern.tampe@med.uni-goettingen.de (B.T.)

**Keywords:** sarcoidosis, Löfgren’s syndrome, vaccination, SARS-CoV-2, COVID-19

## Abstract

Sarcoidosis can present as an acute form or take a chronic course. One of the acute presentations is Löfgren’s syndrome (LS), consisting of the symptom triad of bilateral hilar lymphadenopathy, erythema nodosum, and ankle periarthritis. In addition, there are occasional reports of sarcoid-like reactions following drug exposures. Nevertheless, reports of sarcoidosis or LS after vaccination have not been published. Here, we report two cases of *de novo* LS in a temporal association with different vaccines against the new coronavirus SARS-CoV-2. One patient developed the first symptoms three days after the second vaccination (first vaccination ChadOx-1, Astra Zeneca; second vaccination CX-024414, Moderna); in the second patient, symptoms started 28 days after the first vaccination (ChadOx-1, Astra Zeneca). Both patients eventually required treatment with glucocorticoids. Both patients achieved clinical improvement with treatment. In conclusion, we report the first two cases of LS shortly after SARS-CoV-2 vaccination.

## 1. Introduction

National vaccination programs have been employed to control the ongoing SARS-CoV-2 pandemic and reduce the number of subsequent COVID-19 cases. This is especially relevant in light of emerging variants of concern throughout the world. In Europe, four vaccines have been licensed by the European Medicines Agency. However, they differ by their mechanism of antibody induction (adenoviral or messenger RNA-based [mRNA]) [1]. After reports of adverse events occurring after vaccination ranging from relatively mild cutaneous reactions [2] to severe and potentially fatal conditions, including vaccine-induced thrombocytopenia and thrombosis (VITT) [3], people may be hesitant to receive vaccines. Willingness to receive vaccines against SARS-CoV-2 varied depending on the studied population and ranged from ~70% in a general UK population to over 90% in German healthcare workers [4]. Vaccine hesitancy may be influenced by sociodemographic factors, such as educational level, income, or geographic region [5,6]. By ongoing surveillance and evaluation of adverse event reporting, health authorities nevertheless recommend continuing vaccination programs to achieve herd immunity [7]. With an increasing number of vaccinated people, the more likely is the emergence of adverse reactions. Therefore, it is crucial to report adverse events in temporal association with SARS-CoV-2 vaccination and better understand their management. Here, for the first time, we report the occurrence of Löfgren’s syndrome (LS) in a temporal association with SARS-CoV-2 vaccination in two patients.

## 2. Case Reports

We report two cases of typical LS shortly after SARS-CoV-2 vaccination. Both patients presented to the emergency department (ED) of our center.

The first patient, a 21-year-old Caucasian woman, reported a skin rash three weeks after her second SARS-CoV-2 vaccination. Her past medical history was unremarkable. Erythema nodosa (EN) was present on the palms of both hands, which had started three days after the vaccination and had spread to the face, forearms, and lower legs over the next two weeks. She had received the adenoviral vector vaccine (ChAdOx1, Vaxzevria, AstraZeneca, Cambridge, UK) as the first dose 12 weeks earlier; the second vaccination was performed with an mRNA-based vaccine (CX-024414, Spikevax, Moderna, Inc., Cambridge, MA, USA). This heterologous schedule was chosen based on the German national recommendation that persons younger than 60 who had received ChAdOx1 as the first dose should receive an mRNA-based vaccine as the second dose. In addition to skin findings, immobilizing joint pain in both ankles appeared 18 days after CX-024414. She denied fever or dyspnea; a nasopharyngeal polymerase chain reaction (PCR) test for SARS-CoV-2 RNA was negative. On physical examination, both ankles were swollen and tender; otherwise, the physical examination was normal. Musculoskeletal (MSK) ultrasound showed periarthritis with edema and tendovaginitis of the surrounding tendon sheaths. Laboratory analysis revealed an elevated C-reactive protein (CRP) of 28 mg/L (normal range [NR] < 5 mg/L) and a negative result for procalcitonin. Other laboratory values, including angiotensin-converting enzyme (ACE), were within the normal range. Chest radiography (CXR) was unremarkable. Additional workup did not reveal any signs of other systemic sarcoidosis manifestations. She was treated with ibuprofen at a dose of 600 mg up to three times daily as an outpatient. After 10 days of treatment, she reported persistence of arthralgia. Therapy with glucocorticoids (GCs) was initiated (20 mg/d of prednisolone with a tapering schedule), and she noticed an improvement of symptoms. At follow-up three months after her initial presentation, the patient remained in clinical remission without any relapse of arthritis or skin manifestations.

The second patient, a 27-year-old Caucasian man without a significant past medical history, presented with increasing ankle swelling and pain for three weeks to the ED. One week earlier, he noticed an extensive reddish-violaceous rash on both lower legs. In addition, he reported undulating febrile temperatures and fatigue. Seven weeks before ED presentation, he had received the first SARS-CoV-2 vaccination (ChAdOx1, Vaxzevria, AstraZeneca) without any relevant side effects. He had painful bilateral ankle swelling with periarticular inflammation on MSK ultrasound examination. In addition to an extensive rash on the legs (Figure 1A), additional ENs were present on the forearms and elbows. The physical examination revealed no other relevant findings. The erythrocyte sedimentation rate (ESR) was markedly elevated (80 mm in the first hour [NR < 15 mm]), the CRP was 220 mg/L. The soluble interleukin-2 receptor (sIL2-R) was slightly elevated (854 U/mL, [NR 223-710 U/mL]), ACE levels were normal. Other laboratory analyses were normal. Due to elevated D-dimers, a chest computed tomography (CT) was performed with evidence of a segmental pulmonary artery embolism in the left upper segment artery. In addition, there were bilateral hilar and mediastinal lymphadenopathy and fine nodular parenchymal alterations consistent with sarcoidosis (Figure 1B). The patient was hospitalized and treated with prednisolone at a dose of 20 mg per day and oral anticoagulation. Further workup did not reveal any other sarcoidosis-related organ manifestation. A lymph node biopsy was not performed, given his typical presentation of LS. His symptoms resolved quickly, and he was discharged after four days. At follow-up four months after his first presentation, no new or relapsing symptoms have occurred.

Figure 2 summarizes the disease course of both patients in temporal relation to vaccine administration.

## 3. Discussion

To our knowledge, these are the first reports of LS following vaccination against SARS-CoV-2. Löfgren’s syndrome is characterized by bilateral hilar lymphadenopathy, ankle periarthritis, and erythema nodosum. In its classic form, most experts and current recommendations disfavor histologic confirmation [8,9]. In addition, dyspnea, fatigue, and fever can occur. An increased incidence in spring has been reported [10]. Patients positive for HLA DRB1*03 usually have resolving disease within two years [11]. Symptomatic therapy with non-steroidal anti-inflammatory drugs is often sufficient. If symptoms persist, GC therapy may be required [12].

General adverse events secondary to SARS-CoV-2 vaccines are relatively common among patients with known autoimmune disease (AD); these frequently include injection site reactions, fever, chills, and fatigue [13]. However, these are not more common than in patients without ADs [13]. Furthermore, episodes of AD flares following SARS-CoV-2 vaccination have been described, usually within a maximum of one week after vaccination [14,15]. The new onset of rheumatic conditions has been reported occasionally, including one case of neurosarcoidosis [14]. While lymphadenopathy after vaccination has been observed, no mediastinal or hilar lymphadenopathy has been reported [16]. Although the risk for developing autoimmune diseases secondary to SARS-CoV-2 vaccines is low, we think that a systematic reporting of these events is mandatory. This is especially important in younger patients with a lower mortality risk related to COVID-19 [17].

We cannot completely rule out a coincident seasonal effect of the occurrence of LS in our cases, which occurred in late spring/early summer. Nevertheless, the onset of symptoms between vaccination and the first symptoms is similar to other reports ranging from a few days to four weeks after vaccination [14]. This suggests an immunologic reaction at least triggered by the vaccination. As a limitation of our report, it has to be noted that we have not determined the human leukocyte antigen (HLA) status in our patients. However, it is known that HLA-DRB1*03 is associated with LS and generally confers a good prognosis in terms of recurrence risk [11]. In addition, we have not determined anti-SARS-CoV-2 antibody levels after vaccination because we did not expect a diminished vaccine response, as neither of the patients was immunocompromised at the time of vaccination. Further, neither patient requires continued immunosuppression and, therefore, testing for antibodies after vaccination is not routinely recommended as per German national recommendations [18].

A literature search revealed one additional case of axillary and pulmonary lymphadenopathy associated with sarcoidosis after vaccination against SARS-CoV-2 [19]. However, we were unable to find other cases of LS associated with the vaccination against SARS-CoV-2. Still, reports of sarcoidosis after vaccination exist for different types of vaccines, which are summarized in Table 1.

The occurrence of sarcoidosis after vaccination has been observed for many years. A causal relationship has been a matter of discussion for many years: Early reports from the 1950s and onward have described the development of sarcoidosis after administering the Bacillus Calmette-Guérin (BCG) vaccine [23], which has fueled the hypothesis that sarcoidosis is a result of mycobacterial infection. Direct evidence of mycobacteria in sarcoid lesions has never been demonstrated convincingly [32,33]. Nevertheless, responses to mycobacterial antigens are present in T cells isolated from a bronchoalveolar fluid in sarcoidosis patients [34,35]. Other vaccines that have been described in association with the development of sarcoidosis include the varicella-zoster virus vaccine (Shingrix^®^) [21] and vaccination against seasonal influenza [20] with varying manifestations, including granulomatous uveitis. Complete or near-complete remission is the rule in all these cases, and chronic treatment is usually not required.

Leading international sarcoidosis experts have formulated recommendations for the management of vaccinations and immunosuppressive therapies to protect this vulnerable patient population [36,37], and the rare occurrence of adverse immunological reactions, such as reported here, should not lead to an increased fear of vaccination in general.

## 4. Conclusions

In summary, we report the first two cases of LS starting three days and four weeks after vaccination, respectively. Therefore, we hypothesize that LS may occur as *de novo* immunological reaction to SARS-CoV-2 vaccines, irrespective of their mode of action. Management required GC treatment, but a clinical improvement was achieved quickly in both patients and is reassuring for the treating physicians and affected patients.

## Figures and Tables

**Figure 1 vaccines-09-01313-f001:**
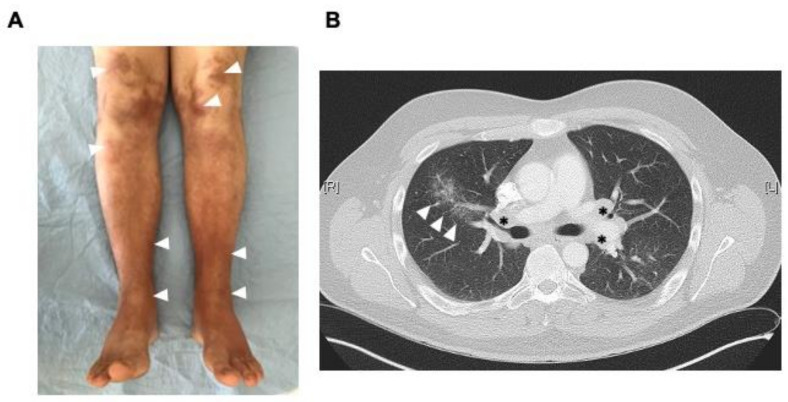
Clinical and radiographic findings in patient #2. (**A**). Extensive rash with erythema nodosa on both legs (arrowheads). Ankles were swollen and tender. (**B**). Chest computed tomography showing interstitial pulmonary infiltrates (arrowheads) and lymphadenopathy (asterisks).

**Figure 2 vaccines-09-01313-f002:**
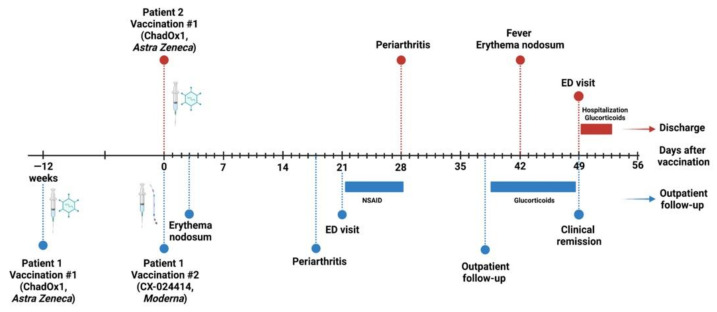
Time course of Löfgren’s syndrome in relation to vaccination in both patients. Created with BioRender.com. Abbreviations: ED, emergency department; NSAID, non-steroidal anti-inflammatory drugs.

**Table 1 vaccines-09-01313-t001:** Summary of findings of sarcoidosis cases after prophylactic immunization.

Type of Vaccine	Timing of Onset	Population	Clinical Manifestations	Laboratory Manifestations	Outcome	Reference
SARS-CoV-2 (BNT162b2, Pfizer/BioNTech)	One day	Adult	Axillary and mediastinal lymphadenopathy	Not reported	Not reported	[19]
Influenza	Two months	Adult	Granulomatous uveitis, Tattoo sarcoidosis, pulmonary lymphadenopathy	ACE elevation	Near-complete remission with a need for topical treatment	[20]
Varicella-zoster (Shingrix)	Four days	Adult	Granulomatous uveitis Bilateral hilar and mediastinal adenopathy	ACE elevation Hypercalcemia Positive RF	Remission with local GC	[21]
BCG	1–84 months	Adult and pediatric	NOD2-associated Blau syndrome with skin, bone. Lung findings and mediastinal lymphadenopathy, EN	ESR elevation in some cases	Spontaneous resolution in most cases	[22,23,24,25,26,27,28,29,30,31]

Abbreviations: ACE, angiotensin-converting enzyme; BCG, Bacille de Calmette et Guérin; ESR, erythrocyte sedimentation rate; GC, glucocorticoids; RF, rheumatoid factor.

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
