# Peer review of "First Report of Two Cases of Löfgren’s Syndrome after SARS-CoV-2 Vaccination-Coincidence or Causality?"

_vaccines, 2021, doi:10.3390/vaccines9111313_

Round 1
Reviewer 1 Report
The paper is well presented and conclusions are clearly supported. One comment, that could be of interest in the discussion of the paper. The young age of the patients also raises ethical questions regarding the advisability of vaccinating the younger age groups in the face of a very low risk of severe disease and a significant increase in possible side effects in the short but also in the long term. The triggering of autoimmune diseases is linked to vaccinations, an ecstatic mass vaccination that does not take into account the risk / benefit ratio by risk group could lead to a very significant increase in the share of people affected by these diseases with an important impact on health management problems.
Author Response
“The paper is well presented and conclusions are clearly supported.”
R/ Thank you for the positive assessment relating to our paper on Vaccine-associated Löfgren’s syndrome.
“The young age of the patients also raises ethical questions regarding the advisability of vaccinating the younger age groups in the face of a very low risk of severe disease and a significant increase in possible side effects in the short but also in the long term. The triggering of autoimmune diseases is linked to vaccinations, an ecstatic mass vaccination that does not take into account the risk / benefit ratio by risk group could lead to a very significant increase in the share of people affected by these diseases with an important impact on health management problems.”
R/ thank you for this comment. We have amended the discussion section accordingly. We think that these are rare events and that the benefits far outweigh this small risk. Nevertheless, we agree that autoimmune and other reactions in association with these relatively novel types of vaccines need to be reported rigorously. We have added a reference (now reference #17) that supports the notion of low death rates among younger patients in Germany.
Reviewer 2 Report
The Authors report the two cases of Löfgren’s syndrome, shortly after vaccination against SARS-CoV-2.
The manuscript is of interest, especially in this very time and it is well written.
The Authors should be congratulated for trying to shed new light on Löfgren’s syndrome. Some comments should be warranted:
Major points:
- The main limitation of the study is the short follow-up of the patients, I would suggest to implement follow-up to improve the manuscript.
Further, it might be of interest to test the patients for immunoglobulins and Ig anti-SARS-CoV-2.
- Löfgren’s syndrome is strongly associated with the HLA-DRB1*03 haplotype. Were the two patients tested?
Minor Points:
- Cardiac or neurologic involvement even tough are rare in Löfgren’s syndrome should be excluded, did the Authors checked for systemic involvement?
- I assume that the previous medical history of the patients was silent? I suggest to clarify it in the manuscript to improve readability.
- Case 1. Athralgia improved after steroid treatment, however how long was the follow-up?
- Case 1. Did the skin manifestations also improved with steroid treatment or it followed a different disease course?
- Löfgren’s syndrome is reported to be more frequent in certain ethnic groups. Could the Authors please provide details of the patients?
- Case 2. The Authors state “several weeks before ED presentation”, please clarify how many weeks.
- Case 2. A lymphnode biopsy was performed to confirm the diagnostic hypothesis?
- Case 2. Is there a follow-up of the patient? Did the Authors performed a follow-up CT?
- Case 1&2. Did the Authors tested the patients for angiotensin converting enzyme?
Author Response
The Authors report the two cases of Löfgren’s syndrome, shortly after vaccination against SARS-CoV-2. The manuscript is of interest, especially in this very time and it is well written.The Authors should be congratulated for trying to shed new light on Löfgren’s syndrome. Some comments should be warranted:
R/ Thank you for your positive comments related to our report.
Major points:
- The main limitation of the study is the short follow-up of the patients, I would suggest to implement follow-up to improve the manuscript.
R/ we have now four and three months of follow-up period for the patients. They are still in remission after treatment with GC. We have amended this in the case descriptions.
- Further, it might be of interest to test the patients for immunoglobulins and Ig anti-SARS-CoV-2.
R/ as per local and national recommendations, we have not done testing for immunoglobulins and Ig anti-SARS-Cov-2 in these patients, because, at the time of writing, a recommendation concerning booster vaccinations was lacking. Now, we also would not derive any clinical consequence from antibody testing in this setting as both patients do not require continued immunosuppression. We added a reference (now ref #18) to the discussion section to explain why we did not do it.
- Löfgren’s syndrome is strongly associated with the HLA-DRB1*03 haplotype. Were the two patients tested?
R/ Unfortunately, we have not tested them for HLA-DRB1*03 as this is a genetic investigation requiring separate consent. We agree, however, that this would be very interesting. We have already stated this as limitation in the discussion section (page 4, line 118)
Minor Points:
- Cardiac or neurologic involvement even tough are rare in Löfgren’s syndrome should be excluded, did the Authors checked for systemic involvement?
R/ both patients were worked up for systemic involvement, which was negative. We added this to the case descriptions.
- I assume that the previous medical history of the patients was silent? I suggest to clarify it in the manuscript to improve readability.
R/ yes, no chronic conditions in both patients. We amended this in the case descriptions.
- Case 1. Athralgia improved after steroid treatment, however how long was the follow-up?
R/ we have not observed relapse of arthralgias. We have now four months of follow up.
- Case 1. Did the skin manifestations also improved with steroid treatment or it followed a different disease course?
R/ also no relapse of skin manifestations at follow-up.
- Löfgren’s syndrome is reported to be more frequent in certain ethnic groups. Could the Authors please provide details of the patients?
R/ we added that both were of Caucasian ancestry.
- Case 2. The Authors state “several weeks before ED presentation”, please clarify how many weeks.
R/ this information has been given already. Page 2, line 67 reads “seven weeks before ED presentation”.
- Case 2. A lymphnode biopsy was performed to confirm the diagnostic hypothesis?
R/ a lymph node biopsy was not done, as the presentation was that of classic LS, in which case most sarcoidosis experts and current recommendations disfavor LN sampling (ref #8).
- Case 2. Is there a follow-up of the patient? Did the Authors performed a follow-up CT?
R/ we followed the patient for four months now. No repeat CT was performed as the clinical presentation did not suggest persistent lung disease.
- Case 1&2. Did the Authors tested the patients for angiotensin converting enzyme?
R/ Yes. Testing was performed and revealed negative results. This is now reported in the case descriptions.
Reviewer 3 Report
This is a very concise and well written paper. A few details that could add significantly to the paper, since so many factors has already been associated with adverse outcomes to both infection and vaccination, are the following characteristics of the patients:
Race/ethnicity (assumed German but not stated)
Personal or familial history of autoimmune disease
BMI or other indicators of obesity (height and weight, % body fat)
It would also be worth noting, if available, why the first patient chose to get AZ then Moderna. Perhaps this was not necessarily planned.
Author Response
- This is a very concise and well written paper.
R/ thank you for your positive assessment of our paper.
- A few details that could add significantly to the paper, since so many factors has already been associated with adverse outcomes to both infection and vaccination, are the following characteristics of the patients:
- Race/ethnicity (assumed German but not stated)
R/ we amended that both patients were of Caucasian ethnicity to the case descriptions.
- Personal or familial history of autoimmune disease
R/ the case descriptions now include that the past medical history was negative in both patients.
- BMI or other indicators of obesity (height and weight, % body fat)
R/ neither patient was obese. We added to the case description, that, apart from the reported abnormalities, the physical examination was normal.
- It would also be worth noting, if available, why the first patient chose to get AZ then Moderna. Perhaps this was not necessarily planned.
R/ we added a comment to the case description of patient #1 that this was based on the national vaccination recommendations in Germany that people under 60 years of age should not receive AZ as a second dose after the occurrence of VITT, especially in younger women.
Round 2
Reviewer 2 Report
The Authors have addressed all the concerns.